# Understanding physician behaviour in the 6–8 weeks hip check in primary care: a qualitative study using the COM-B

Angel Chater,[1] Sarah Milton,[2] Judith Green,[2] Gill Gilworth,[2] Andreas Roposch [3,4]

[1]Department of Sport Science and Physical, University of Bedfordshire, Luton, UK
[2]Department of Population Health Sciences, King's College London, London, UK
[3]Institute of Child Health, University College London, London, UK
[4]Department of Orthopaedic Surgery, Great Ormond Street Hospital for Children, London, UK

**Correspondence to**
Professor Andreas Roposch;
a.roposch@ucl.ac.uk

## ABSTRACT

**Objectives** A compulsory hip check is performed on an infant at 6–8 weeks in primary care for the detection of developmental dysplasia of the hip (DDH). Missed diagnoses and infants incorrectly labelled with DDH remain an important problem. The nature of physician behaviour as a likely source of this problem has not been explored. The aims of this study were to make a behavioural diagnosis of general practitioners (GPs) who perform these hip checks, and identify potential behavioural change techniques that could make the hip checks more effective.

**Design** Qualitative study with in-depth semistructured interviews of 6–8 weeks checks. We used the Capability, Opportunity, Motivation and Behaviour model in making a behavioural diagnosis and elicited factors that can be linked to improving the assessment.

**Setting** Primary care.

**Participants** 17 GPs (15 female) who had between 5 and 34 years of work experience were interviewed.

**Results** Capability related to knowledge of evidence-based criteria and skill to identify DDH were important behavioural factors. Both physical (clinic time and space) and social (practice norms), opportunity were essential for optimal behaviour. Furthermore, motivation related to the importance of the 6–8 weeks check and confidence to perform the check and refer appropriately were identified in the behavioural diagnosis.

**Conclusion** Aspects of capability, opportunity and motivation affect GPs' diagnosis and referral behaviours in relation to DDH. The findings from this work extend current knowledge and will inform the development of an intervention aimed at improving the diagnosis of DDH.

## INTRODUCTION

Developmental dysplasia of the hip (DDH) is characterised by varying displacement of the proximal femur from the acetabulum with associated acetabular dysplasia. Dislocation occurs in 1–2/1000 infants per year but milder forms occur in 40–60/1000 newborns.[1] Because early recognition of disease is associated with better outcomes, the compulsory Newborn and Infant Physical Examination (NIPE) dictates that all infants are examined for the presence of DDH at 6–8 weeks in primary care. If diagnosed this early, splinting of the hips is successful in 85%

of cases.[2 3] Later diagnosed DDH will usually require prolonged and invasive treatment, and is associated with long-term disability.[4]

### Strengths and limitations of this study

► This was the first study on the diagnosis of developmental dysplasia of the hip to focus specifically on the behaviour and behavioural change of diagnosticians.
► The study points to the drivers of behaviour and behavioural change techniques that are required in order to improve the 6–8 weeks hip check.
► While there was balanced representation of diagnosticians based in diverse socioeconomic areas, practice setting and professional experience the study was confined to London.

NIPE is typically conducted by a general practitioner (GP) and includes screening for disorders of the heart, eyes, testes and hips. The NIPE handbook[5] provides general guidance on the conduct of the examination. For example, it outlines how to undertake the hip examination, which risk factors to record, how to manage parental concerns and how to proceed if DDH is diagnosed or suspected. The effectiveness of the 6–8 weeks hip check remains controversial.[6] Late diagnosed DDH remains a problem as are the high number of infants labelled incorrectly as DDH and referred to secondary care. A recent 15-year review of the programme,[6] concluded the –8 weeks hip check performs poorly. Several potential explanations exist. The signs of hip instability (detected by the so-called Barlow and Ortolani clinical tests) are resolving at the age of 6–8 weeks when the other explicit clinical test, (ie, limited hip abduction), has not yet developed.[7] Other potential barriers include a lack of precisely formulated instructions about the conduct of the hip examination; variability among GPs in adhering to explicit referral criteria; a lack of consistency in terms of time and space in performing the

6–8 weeks check consultations; or a lack of experience in examining the hips of infants.

In prior research, we linked personal beliefs, geography and training backgrounds of physicians with variability in the diagnosis of DDH.[8 9] Interventions aimed at reducing variability could potentially lead to more effective diagnosis. Behavioural change theory has been used successfully in changing physician behaviour and linked with improved outcomes.[10] One such approach is the Behaviour Change Wheel.[11] It is based on three layers that are to be considered when supporting behavioural change. Its first layer, Capability, Opportunity, Motivation and Behaviour (COM-B) focuses on the determinants of behaviour, thus enabling a 'behavioural diagnosis'. A behavioural diagnosis is the assessment of influences on a desired behaviour; it includes consideration of factors that may either impede or facilitate behaviour. A behavioural diagnosis enables the identification of intervention strategies and behavioural change techniques, which, in turn, could improve outcomes by facilitating behavioural change.

The aims of this study were to make a behavioural diagnosis of GPs who routinely conduct 6–8 weeks hip checks, and to identify determinants of behaviour based on COM-B that have the potential to make the checks more effective.

## METHODS

### Design

A qualitative investigation using semistructured interviews.

### Participants and setting

Seventeen GPs (15 females) who routinely conduct the compulsory 6–8 weeks check across 16 surgeries in London, England, participated. We covered a variety of areas in terms of relative diversity, demographics and levels of affluence/deprivation. Some GPs had dedicated weekly times for the 6–8 weeks check, whereas others conducted these across the week in regular consultation times. Consultations took between 15 and 28 min.

### Materials

A semistructured interview schedule was designed, framed to the constructs of COM-B (table 1). This ensured that questions were asked in relation to Capability, both

**Table 1** Questions and prompts used in the interview schedule and their relationship to COM-B constructs

| Semistructured interview questions | Prompts | COM-B construct |
|---|---|---|
| What happens during the 6 weeks check? (identify 'usual care') | Order of activities (mother/baby), priorities | Capability |
| | Physicality of the space/arrangement of the room for example, location of computer in comparison to GP/mother/baby | Opportunity |
| | Cases that have been particularly challenging | Motivation |
| How would you examine for hip dysplasia in infants? | What are you looking/feeling for? for example, 'click', other risk factors if not mentioned | Capability |
| | Probe feelings and beliefs (eg, confidence) around doing this | Motivation |
| Describe your experience of hip dysplasia | Any experience? Experience of diagnosis at a later age? | Capability |
| | Knowledge of others' experience of diagnosis or misdiagnosis? | Capability |
| What happens next if diagnosed? | Referral behaviour | Behaviour |
| How do you know what to do during the 6-week check and how to examine for hip dysplasia | Training? When was this? What did this include? | Capability |
| | Any events/meetings/networks to update knowledge/training? | Opportunity Capability |
| | Is this individually led? | Motivation |
| What guidance do you use for making decisions? What would stop you from using them? | In the 6 weeks check? In general? | Opportunity |
| | What format (computer, paper, poster, etc)? | Opportunity |
| | What are the advantages/disadvantages of different formats? How helpful are they? | Motivation |
| Have you experienced the introduction of a decision aid in your practice? | How was this experienced? | Behaviour |
| What would help with the potential challenges of diagnosing and referring on for hip dysplasia? | Any suggestions for a possible intervention? | Behaviour |

COM-B, Capability, Opportunity, Motivation and Behaviour; GP, general practitioner.

physical (eg, skills) and psychological (eg, knowledge); Opportunity, both social (eg, norms of practice) and physical (eg, time/space) and Motivation, both reflective (eg, confidence and intention) and automatic (eg, driven by emotion or habit) that may all influence GP's behaviour at the 6–8 weeks check.

Written informed consent was obtained from all participants.

## Procedure

The study ran from September 2018 to August 2019. Experienced qualitative researchers from King's College London (SM, GG) conducted individual, face-to-face, semi-structured interviews with all participants. Researchers had no previous relationship or interactions with participants. These were audiorecorded and transcribed verbatim for analysis. Mean interview time was 42 min (range 20–75 min). As part of the interviews, each participant had the chance to look at and discuss a checklist that we designed based on prior research in order to aid referral decisions for suspected DDH.[4] This step was done to (1) visualise certain details of the 6–8 weeks hip check (and thus stimulate an interviewee's views on the components of the hip examination as outlined in the NIPE handbook[5] and (2) to explore the impact of structure in the conduct of the 6–8 weeks hip check. In brief, the 'checklist' includes seven explicit and binary standardised diagnostic criteria for DDH.[4 12] These can readily be elicited in the consulting room (eg, Ortolani test, Barlow test, leg length differences) and are part of the NIPE handbook.[5] The checklist makes a recommendation about circumstances in which to refer an infant with suspected DDH to secondary care. The efficacy of this checklist will be assessed alongside audiovisual instructions in a future randomised clinical trial.[13] Qualitative research undertaken by the authors of this paper has informed the design of the checklist, as well as procedures to be undertaken in the trial. Further qualitative research will be undertaken in an ongoing process evaluation of the trial.

## Analysis

Transcripts were initially coded by SM/GG using computer software (NVivo), taking an initial inductive approach which drew on elements of the 'constant comparative method' to identify common themes, variation and deviant cases in order to investigate the dataset fully.[14] Initial codes and identified themes were then discussed and reviewed with two other experienced qualitative researchers (JG and AC). Parameters were set to consider saturation[15] (ie, when interview data were categorised within the developing coding frame with no additional new codes or themes identified), and this was achieved, through discussion with SM, GG, JG and AC, after 10 interviews. Following agreement, SM and AC deductively mapped themes to the appropriate constructs of COM-B using a similar approach to past research.[16]

## Patient and public involvement

We developed the grant proposal, for which this study is part of, with input from GPs, carers of children with DDH and the founding director of 'Steps', a charity supporting patients with lower limb disorders. Our established patient and public involvement group periodically review and comment on study documents and findings, and have supported this current study with insight into the 6–8 weeks check process.

## RESULTS

Initial inductive analysis identified the following overarching themes that influence behaviours related to the 6–8 weeks check: 'Training in paediatrics'; 'Knowing hip examination terminology and NIPE guidance'; 'Examination skills'; 'Experience in practice'; 'Designated baby clinics and time available'; 'Interaction with computer system'; 'Uncertainty in the physical hip examination'; 'Beliefs about hip examination as a priority among other examinations'; 'Meaning and understanding of referral'; 'Low threshold to refer'; and 'Referrals need to be justified based on explicit criteria'.

From this initial analysis, we deductively identified clear Capability, Opportunity and Motivation influences on GP behaviour in relation to DDH, which provided evidence for components which could be targeted in future interventions to improve referral pathways. This COM-B analysis is the focus of the current paper.

### Capability

All GPs mentioned that they examine hips during the 6–8 weeks check as part of other examinations. Many questioned whether they, or GPs in general, were performing the hip examination 'correctly' or 'effectively', highlighting a potential issue with their knowledge (psychological capability) and skills (physical capability) in the procedure.

> If you don't do anything else in a newborn examination you would know to check the hips […] so I don't understand how people wouldn't do it but I understand how people would do it incorrectly. (GP 1)

> I suppose the trouble is that a lot of these are 'feeling things' and no matter how much you tick the boxes, it's a matter of whether you recognise it when you feel it. The abduction is fairly clear, but even that is partly a feeling… (GP 4)

Uncertainty was related to the fact that while the hip examination for DDH is taught during medical school, for those who do not elect to undertake paediatric training thereafter, this may have been many years ago. In addition, training to date had generally not provided them with experience of the 'feel' of an abnormal hip. GPs recollected learning to do hip examinations on dolls, which did not replicate the 'clunk' of an abnormal hip: this was something they just had to learn from experience. However, given that abnormal hips are infrequently

seen in primary care, few had felt a 'clunk', leading to questioning of a skill linked to physical capability.

> I've never felt it in real life or heard it in real life… I know what I'm feeling for but I've never felt it or heard it… I think the closest I've felt to it is like a dislocated joint in an adult in A&E. (GP 3)

> In the back of your mind, you are thinking, well, how would I know if I found that there was DDH? You hear about the clunk and the clicks and things, but the thing is if you've never come across it, is it because you're not looking hard enough or you weren't looking for the signs? (GP16)

Informal and experiential learning of the manipulations needed in the examination also meant many GPs were unsure of the names of the Ortolani and Barlow tests (as specified in the NIPE handbook), or which was for a dislocated hip that could be reduced, and which for a hip that can be dislocated. In terms of the checklist, which had separate items listed for these tests, this uncertainty highlighted a psychological capability in relation to knowledge:

> I never remember which is Ortolani and which is Barlow actually, I think one is out and one is in. (GP 4)

> I just refer to them [the Ortolani and Barlow tests] as 'Barlani', you do it all at the same time don't you? (GP 10)

Commenting on the checklist-led GPs to acknowledge that not all items were habitually covered, and that some items were no longer considered useful (such as asymmetric hip creases) were still used:

> I think there might have been a hip' crease asymmetry at some point [as a reason for referral] (GP2)

> …say if it was breech delivery and they hadn't had a hip scan organised in hospital already, then I would refer for that. Or if, I guess, their family history, although saying that, I don't ask specifically about that. (GP 7).

In general, the perceived future relevance of a checklist was rated by the GPs in relation to how they were orientated to the role of 'evidence-based practice' in informing their behaviour.

> Really helpful, just because it's very black and white. If you have a decision tool it's very useful for some who like me is quite junior, because then its written on paper, like you can follow the flowchart and show how you made, why you made that decision. (GP 2)

> Anything that helps pick things up is fantastic. General practice is so broad and general as you know. It's a two- minute examination and so reminders about all the risk factors [would be helpful]. (GP 5)

## Opportunity

The time available for infant checks varied across GP practices, affecting the physical opportunity for the DDH examination and for enacting appropriate referrals. Where there were specific times and days set aside for a 'baby clinic', these were typically longer appointments than the ten minutes allocated to most primary care appointments. The time taken for the infant check varied between 15 and 28 min. Longer appointments were reported to provide better opportunities to undertake the examinations, and identify parental concerns, particularly given the practical challenges of examining young infants:

> The baby checks we used to have to do in 10 [minutes] but I've asked, you know, it needs to be 15 or 20. Sometimes I get 20, sometimes I don't, but then it just eats into the rest of the clinic time because you have to go through all of these things. (GP 1)

> So it's fifteen minutes for the mum and fifteen minutes for the baby […] When it's straightforward then I think that's adequate…often [the infants] come in sleeping and I often will try and listen to their heart, or if they're lying quietly awake I'll try and look in the back of their eyes. So the two things I try and get out of the way… because that's the bits I can't do when they're crying. (GP 12)

Whether or not the GP practice had designated baby clinics was related to the size of the practice, with the smaller practices reporting carrying out baby checks 'as and when' needed. Those without designated baby clinics may have less time allocated for the consultation, and may be less likely to remember to use a checklist, as infants may be seen for example once every few weeks. The physical environment where the checks occurred also varied and individuals could be moving from computer to baby and back to ensure everything has been covered. Prompts to use a checklist, covering each aspect of the check, were seen as something that could be helpful.

> No, not at my practice [we don't have specific times of the week for baby checks]. We just, um, as it comes. (GP 1)

> And I guess it would be nice for there to be something, like, that could pop [up], if you clicked like a wrong, or an abnormal result, would pop up with like what to do … sometimes I find that you do it, you think you've finished and then you go to fill in the Red Book or look on the computer and you're like, "Oh, I have to do something, I have to go back and do something' (GP 3)

However, some GPs identified potential difficulties with a computerised checklist, if it interrupted the flow of the consultation, which could be a highly emotive and vulnerable time for the parents. This highlights issues with the social opportunity to engage in effective decision making in relation to the hip examination and referral behaviour, in the context of a check, which is about more than just DDH:

Sometimes the first time you meet a mum after six weeks, and it's so much, that's your chance to really ask about mood and coping and emotion and talk about sex for the first time and contraceptives. Huge issues. More and more I'm sort of driven slightly away from checklisty type things because it erm… gets you at the computer. (GP 2)

In some practices, there was evidence of social opportunity enhancing referral behaviours, as these were discussed as joint decisions by a practice, rather than an individual GP. Such discussions would provide opportunities to develop skills in identifying appropriate grounds for referral with colleagues:

Every referral is essentially discussed with another GP or another doctor. It's very much a team environment, so you work very closely with other GPs. We all get together in the coffee room at least twice, but usually three times during the day. (GP 7)

## Motivation

The other checks that have to be conducted as part of NIPE shaped GPs' motivation in relation to the perceived relative importance of the hip examination:

I guess it's taking into context that examining the hips, although it's an important part of baby checks, so much of it is an important part of the baby check and the mum check, and no, we don't want to miss a hip, but, equally we don't want to miss, you know, an eye or miss a femoral pulse. (GP 7)

Seeing the hip examination as a priority was also diminished by the fact that abnormal hip joints were rarely seen or felt. By extension, this impacted on reflective motivation, leading some GPs to question whether their time would be used effectively by adding a checklist to an already busy consultation.

You are checking so much, but there are a few things that you would, hopefully you wouldn't forget any of it, but a few things that you're really paying close attention to, so you're paying really, really close attention when feeling the palate for a cleft palate, and when you're listening to the heart for any murmurs, and…checking the red reflex in the eyes and checking the femoral pulses, checking the hips and the leg creases, really, because we don't want to miss it. (GP 2)

I think, you know, how many babies do we see? In the twenty years of examining babies, up to three a week, there's been one case of developmental dysplasia of the hip - and that case was picked up appropriately. With all the things that we have to do, find a quick intervention that makes us better at developing, detecting it, great, but a five min questionnaire, you have to click on a different page to go and do that, vs the effect that's going to have. (GP 7).

However, GPs who had volunteered to try out the checklist during their 6–8 weeks checks found it generally easy to use, and not adding to their workload:

It's just putting it down and making you think this is what you're looking for, I think it's a very useful tool. (GP13)

I don't think it's increased my workload, no, because it's not really changed anything significantly in time. So no, it's of no extra pressure to do it. (GP15)

On referral behaviour, GPs' reflective motivation was rooted in their understanding and practice of referral to secondary care in general and their beliefs about the consequences of their referral. They considered the specific opportunities afforded by local referral pathways, such as whether an ultrasound could be obtained without a referral to an orthopaedic specialist. In the context of a national health system, some were motivated to minimise referrals to reduce public costs:

We definitely feel that if it's something that can be managed in primary care; that we would really try to do everything we could here, so we try to avoid unnecessary referrals. (GP 7)

I'd rather see the baby again myself for a repeat appointment, rather than refer them on, you know. If I was uncertain about something like the hips, I'd see the baby again. Paediatric orthopaedics, their time is really valuable. (GP 5)

Balancing this was an orientation towards treating secondary care as a legitimate source of second opinion, particularly if the referral was simply for an ultrasound, which would inform them better of whether an orthopaedic specialist referral would be needed:

The babies get ultrasounds; it's very easy to organise, they're done within a week or so and they're back and nobody minds a normal result. (GP11)

GPs also considered the scarcity of their own resources, and reported at times referring to avoid patients returning to them if they were uncertain about a diagnosis. Others reflected on the inherent tensions of referral pathways from primary care, given their own generalist skills compared with the expertise of paediatric orthopaedics, which was seen as the legitimate place to reduce uncertainty.

Too many and I'm probably under-referring and too few and I'm over-referring, what's just right? There is no answer, there is no 'just right,' that's what nobody will really come out and say. If that's what they mean, that people waste the time in [hip] clinics by sending them hips that are okay, well that's sort of what they're there for. (GP 4)

Changing examination and referral behaviour may be potentially undermined by underlying drivers of motivation such as the lack of belief of the importance of the

check at 6 weeks (reflective motivation) and the GP's social/professional role and identity (reflective/automatic motivation) around the check. Some GPs believed that, as infants had already been seen by a specialist at the hospital after birth, any hip abnormalities should have been picked up then. The baby check within the GP practice therefore could be perceived as just a repeat check of something that had already been carried out by someone more specialised.

> I'm doing the six or eight week check but a check would have been done when the baby was born as well, so I would have figured that the paediatrician would have done that as well and that would have been picked up. (GP 1)

## DISCUSSION

We sought to elucidate the determinants of behaviour of GPs who routinely conduct the 6–8 weeks hip check and to highlight the factors that might be instrumental in creating an intervention aimed at changing behaviour. Such an intervention could potentially reduce variability in diagnosing DDH in primary care, with the potential for improve clinical outcomes.

Drawing from COM-B[11] there are a number of factors to consider based on the evidence from this study. First, to support change in referral behaviour, it may be useful to provide details of the clinical importance of the 6–8 weeks check in the context that postnatal hip checks are limited in identifying DDH (ie, irreducible dislocations).[6] In relation to Capability, GPs might benefit from improved education and training on the importance of the 6–8 weeks check, the names of the various clinical tests and instructions on how to perform them. While physically feeling the actual 'clunk' of a hip would be the ultimate learning opportunity, perhaps with greater uptake of training courses provided with life-like dolls, a diagnostic aid with an associated video could also target these Capability issues.

For Opportunity, the physical environment impacts on the GP's ability to hold an effective consultation, and it is important that they are given sufficient time to conduct the 6–8 weeks check. Any intervention such as a diagnostic aid should be created to integrate with the current physical environment and clinical practice, to facilitate its use and subsequent behaviour. For example, this could be in relation to using the diagnostic aid on a desktop computer screen, which are used routinely during consultations in order to access electronic patient records. Social Opportunity should also be targeted, with practice-level support encouraged so that referral decisions can be discussed with other GPs, such as during practice meetings. The influence of the relationship with the parents should also be noted, with findings from our previous research highlighting the importance of explaining the examination.[17]

Motivation towards referral behaviour was related to confidence in the ability to make an appropriate referral and the importance of the examination in the first place, given how relatively rare DDH is in primary care. Knowing how frequent referrals from the practice are, and guidance from trainers and senior partners, may also help to build these elements of Motivation in any interventions, using feedback and audit processes and creating supportive norms of practice and social support to build confidence.

We note the limitations of our study. A relatively small number of GPs from one city in the UK participated, and we cannot make claims for the generalisability of any particular barriers or facilitators of behavioural change identified here. For example, there may be other factors to consider outside of a London-based setting. However, our sample size exceeded that suggested for operationalising data saturation for theory-based interview studies[14] and data reached the stopping criteria at 10 interviews, in that no new ideas were being identified, suggesting that it had achieved saturation. This enabled the key aim to be met, which was to identify via a COM-B behavioural diagnosis, what might influence referral behaviour, and how this can inform components of an intervention based on the use of a decision aid that could be embedded within routine practice. It should be highlighted, however, that there may be some practices where the hip examination is undertaken by someone other than the GP, such as Advanced Care Practitioners (nurses, physiotherapists, midwives, community paediatricians, health visitors). It would be important to research these practitioners and compare their results with GPs in a future study.

## CONCLUSION

Using the COM-B framework, we identified aspects of Capability, Opportunity and Motivation that affect GPs' referral behaviours in relation to DDH. The findings from this work will inform the refinement of a decision aid that aims to improve screening for DDH in UK primary care.

**Contributors** AC, JG, SM, AR substantially contributed to the conception and design of the work. AC, SM and GG contributed equally to acquisition and analysis of the data. All authors contributed to the interpretation of data. AC, SM and AR wrote the first draft; all authors revised the draft and approved the final manuscript. AR, JG and AC secured the funding for the work (AR is chief investigator). All authors approved the final version to be published and agree to be accountable for all aspects of the work in ensuring that questions related to the accuracy or integrity of any part of the work are appropriately investigated and resolved.

**Funding** This paper presents independent research funded by the National Institute for Health Research (NIHR) under its Programme Grant for Applied Research funding stream (RP-PG-0616–20006).

**Disclaimer** The views expressed are those of the authors and not necessarily those of the NIHR or the Department of Health and Social Care.

**Competing interests** None declared.

**Patient and public involvement** Patients and/or the public were involved in the design, or conduct, or reporting, or dissemination plans of this research. Refer to the Methods section for further details.

**Patient consent for publication** Not required.

**Ethics approval** King's College London University Ethics Board (MRA-17/18–6433) and Health Research Authority (18/SW/0168) approved this study.

**Provenance and peer review** Not commissioned; externally peer reviewed.

**Data availability statement** Data are available on reasonable request. All data have been handled in accordance with the UK Data Protection Act 2018. Data have been transcribed and coded using computer software NVivo. All raw data are confidential and resides with Kings College London.

**ORCID iD**
Andreas Roposch http://orcid.org/0000-0002-0143-7840

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
