## [Reviewer comments · BMJ Open]

ARTICLE DETAILS

TITLE (PROVISIONAL)	Understanding physician behaviour in the 6- to 8-week hip check in primary care: a qualitative study using the COM-B
AUTHORS	Chater, Angel; Milton, Sarah; Green, Judith; Gilworth, Gill; Roposch, Andreas

VERSION 1 – REVIEW

REVIEWER	Professor Robin W Paton Medical School, UCLAN Royal Blackburn Teaching Hospital, East Lancashire Hospitals NHS Trust, Haslingden Rd., Blackburn UK
REVIEW RETURNED	18-Nov-2020

GENERAL COMMENTS	This is an interesting small qualitative study assessing the behaviour of GPs conducting the 6 to 8 week hip check. A capability, opportunity and motivation model is used. The questions set to the GPs are well constructed and the answers are revealing regarding issues with the Ortolani and Barlow tests, the knowledge and confidence of the GPs. Factors on changing GP behaviour are discussed: reducing variability, education, adequate time, diagnostic aids and improving the effectiveness of the hip examination. The limitations of the study are that it uses a small number of GPs, it is in one city and in many practices the hip examination is increasing being undertaken by non-doctors: Advanced Care practitioners (Nurses, physiotherapists, mid-wives). It would be important to survey these practitioners and compare their results with GPs in a future study. Conclusion: This is the first study in assessing behaviour and behaviour change in the DDH 6 to 8 week hip check. it gives valuable insights into this behaviour and the limitations of this hip check in GPs. A larger study including GPs and other health professionals involved in the 6 to 8 week hip check would be the logical next step in assessing the effectiveness of this controversial clinical hip screening assessment (in conjunction with a clinical study assessing the effectiveness of the test (sensitivity, specificity and PPV)
--

REVIEWER	Debbie Smith The University of Manchester, UK
-----------------	--

	I am currently conducting work on the 6-8 week postnatal check conducted by GPs but focusing more widely on behaviour change so no competing interests but worth mentioning.
REVIEW RETURNED	21-Nov-2020

GENERAL COMMENTS	Thank you for the chance to review this paper. I really enjoyed reading this paper as I believe the 6-8 week check is a really important time for behaviour change. I have not really given the focus of this paper, the hip check, much thought in my consideration of behaviour change at this time so I was very interested to learn of this work. I think this is a great study with a clear and detailed rationale and discussion, I just have a few comments as to where I would like to see a little more detail or clarity in the method and results sections, please see below:  -I am sorry but I was a little confused as to what data was actually included in this paper. The 'abstract' motioned that 17 interviews were conducted and 14 GPs were observed then the 'participants and settings' section mentions the length of the observations. However, other than this I cannot see any mention or inclusion of these observations - were they included in the analysis or did this inform the interview topic guide? Some clarity would be good here. -The 'procedure' section mentions a 'checklist', I assume this is an aid for GPs to use to guide referral which is a great idea (we have designed a similar one more generally for behaviour change!). This is mentioned again in the results as it says that a few participants volunteered to use this. I was not fully sure about the process of this inclusion e.g., were all the GPs given it and advised to use it? Some clarity would be good. Also, this made me wonder whether this should have featured in the study aims if you are asking about the feasibility of this checklist? -The 'method' section talks about 'saturation was achieved' and 'deviant cases' but I felt I was unable to understand these processes and decisions and needed more information to do so. The idea of data saturation has been widely discussed in the literature over the last few years so some of this literature and consideration of this process (which has a grounded theory base) would be good here to show the reader the true picture of what decisions were made around data collection and namely when to cease collection. -I would like to see some more detail in the results section. The analysis plan talked about two steps to analysis: inductive and deductive. This makes sense for the study aim and was outlined well. However, we see little detail and evidence of stage 1 (the development of the inductive themes) in the results and instead see the outcome of the deductive analysis as we see data by COMB component. Again, this detail is good but I had expected to see details of step one and the themes identified there. This made me wonder why this stage was needed, could you have not coded the data deductively without this stage? It maybe that the themes were mentioned here e.g., 'uncertainty' and just needed some highlighting to show the steps. -You mention PPI involvement which is great - what exactly did they do for this study? This detail would be great.
--

VERSION 1 – AUTHOR RESPONSE

Authors' Response to Comments of Reviewers

Reviewer Remarks	Authors' Responses	Text Changes (line number in clean version)
REVIEWER 1		
This is an interesting small qualitative study assessing the behaviour of GPs conducting the 6 to 8 week hip check. A capability, opportunity and motivation model is used. The questions set to the GPs are well constructed and the answers are revealing regarding issues with the Ortolani and Barlow tests, the knowledge and confidence of the GPs.	Thank you for your positive comments.	N/A
Factors on changing GP behaviour are discussed: reducing variability, education, adequate time, diagnostic aids and improving the effectiveness of the hip examination.	Thank you for your review.	N/A
The limitations of the study are that it uses a small number of GPs, it is in one city and in many practices the hip examination is increasing being undertaken by non-doctors: Advanced Care practitioners (Nurses, physiotherapists, mid-wives). It would be important to survey these practitioners and compare their results with GPs in a future study.	Thank you for this comment, and we agree that it would be valuable to expand future research in order to explore and compare the practices of Advanced Care practitioners.	We have added the following sentence to the discussion on line 456: It should be highlighted, however, that there may be some practices where the hip examination is undertaken by someone other than the GP, such as Advanced Care Practitioners (nurses, physiotherapists, midwives, community paediatricians, health visitors). It would be important to

		research these practitioners and compare their results with GPs in a future study.
Conclusion: This is the first study in assessing behaviour and behaviour change in the DDH 6 to 8 week hip check. It gives valuable insights into this behaviour and the limitations of this hip check in GPs. A larger study including GPs and other health professionals involved in the 6 to 8 week hip check would be the logical next step in assessing the effectiveness of this controversial clinical hip screening assessment (in conjunction with a clinical study assessing the effectiveness of the test (sensitivity, specificity and PPV)	Many thanks for your positive feedback. We are now in the early phase of a larger study looking at the aspects that you suggest and the using of a further development checklist from this study.	We have added the following and a citation of the protocol for our clinical trial, line 153: The efficacy of this checklist will be assessed alongside audio-visual instructions in a future randomised clinical trial.¹³ Qualitative research undertaken by the authors of this paper has informed the design of the checklist, as well as procedures to be undertaken in the trial. Further qualitative research will be undertaken in an ongoing process evaluation of the trial. ¹³Roposch A, Warsame K, Chater A, Green J, Hunter R, Wood J, Freemantle N and Nazareth I. Study protocol for evaluation of aid to diagnosis for developmental dysplasia of the hip in general practice: controlled trial randomised by practice. BMJ

		Open 2020;10:e041837.
REVIEWER 2		
Thank you for the chance to review this paper. I really enjoyed reading this paper as I believe the 6-8 week check is a really important time for behaviour change. I have not really given the focus of this paper, the hip check, much thought in my consideration of behaviour change at this time so I was very interested to learn of this work. I think this is a great study with a clear and detailed rationale and discussion, I just have a few comments as to where I would like to see a little more detail or clarity in the method and results sections, please see below:	Thank you for your positive comments.	N/A
I am sorry but I was a little confused as to what data was actually included in this paper. The 'abstract' motioned that 17 interviews were conducted and 14 GPs were observed then the 'participants and settings' section mentions the length of the observations. However, other than this I cannot see any mention or inclusion of these observations - were they included in the analysis or did this inform the interview topic guide? Some clarity would be good here.	Thank you for noting this lack of clarity. The data included in this paper is drawn from the interviews that formed part of a wider study that included observations, with the aim of future analysis that will use NPT: Normalisation Process Theory.	We have edited the paper to reflect this more accurately and removed reference to observations throughout.
The 'procedure' section mentions a 'checklist', I assume this is an aid for GPs to use to guide referral which is a great idea (we have designed a similar one more generally for behaviour change!). This is mentioned again in the results as it says that a few participants volunteered to use this. I was not fully sure about the process of this inclusion e.g., were all the GPs given it and advised to use it? Some clarity would be good.	Thank you for highlighting this, and it is good to know there are other checklists for behaviour change that have been designed. The abstract mentions that the findings from this work will inform the development of an intervention aimed at improving the diagnosis of DDH. We have made this more explicit by providing more detail in the Procedure section of the Methods. We have also clarified that all GPs were given the chance to see the checklist in the interviews to aid the discussions.	We have adapted the paragraph from line 144 to read: As part of the interviews, each participant had the chance to look at and discuss a checklist that we designed based on prior research in order to aid referral decisions for suspected DDH.⁴ This step was done to (i) visualize certain

		details of the 6- to 8-week hip check (and thus stimulate an interviewee's views on the components of the hip examination as outlined in the NIPE handbook⁵ and (ii) to explore the impact of structure in the conduct of the 6- to 8-week hip check.
Also, this made me wonder whether this should this have featured in the study aims if you are asking about the feasibility of this checklist?	We have not included this in the study aims, as the aim of the paper and the data analysis fort his study was to produce a 'behavioural diagnosis' of practitioners undertaking the 6-8 week check. We will publish other papers that will discuss the checklist, intervention and trial more fully.	From line 153 we have added: The efficacy of this checklist will be assessed alongside audio-visual instructions in a future randomised clinical trial.¹³ Qualitative research undertaken by the authors of this paper has informed the design of the checklist, as well as procedures to be undertaken in the trial. Further qualitative research will be undertaken in an ongoing process evaluation of the trial.
The 'method' section talks about 'saturation was achieved' and 'deviant cases' but I felt I was unable to understand these processes and decisions and needed more information to do so. The idea of data saturation has been widely discussed	Thank you for highlighting this. We drew on the constant comparison method, which includes investigating deviant cases so as to confirm/contradict identified themes. To explain 'saturation' we wrote that the interview data produced after 10	We have inserted the reference for decisional methods relating to stopping criteria

in the literature over the last few years so some of this literature and consideration of this process (which has a grounded theory base) would be good here to show the reader the true picture of what decisions were made around data collection and namely when to cease collection.	interviews were included in the analytical framework that we had drawn up by that point i.e. rather than making further, new codes after 10 interviews. This is because after 10 interviews we were confident that rather than 'new' themes, what was identified was further content to add to the themes we already had in place. We based this process on the Francis et al (2010) paper, with guidance on 'stopping criteria' of 10 interviews plus 3 when operationalising data saturation for theory-based interview studies.	¹⁴Francis JJ, Johnston M, Robertson C, Glidewell L, Entwistle V, Eccles MP, Grimshaw JM. What is an adequate sample size? Operationalising data saturation for theory-based interview studies. Psychology and health. 2010 Dec 1;25(10):1229-45. We have reworded the following sections: Line 160: Transcripts were initially coded by SM/GG using computer software (NVivo), taking an initial inductive approach which drew on elements of the 'constant comparative method' to identify common themes, variation, and deviant cases in order to investigate the dataset fully. Initial codes and identified themes were then discussed and reviewed with two other experienced qualitative researchers (JG, AC). Parameters were set to
---	---	---

		consider saturation¹⁴ (that is, when interview data were categorised within the developing coding frame with no additional new codes identified), which was achieved after 10 interviews. Following agreement, SM and AC deductively mapped themes to the appropriate constructs of COM-B using a similar approach to past research.¹⁵ Line 453: ...our sample size exceeded that suggested for operationalising data saturation for theory-based interview studies¹⁴ and data reached the stopping criteria at 10 interviews, in that no new ideas were being identified, suggesting that it had achieved saturation. This enabled the key aim to be met, which was to identify via a COM-B behavioural diagnosis, what might influence referral behaviour, and
--	--	---

		how this can inform components of an intervention based on the use of a decision aid that could be embedded within routine practice.
I would like to see some more detail in the results section. The analysis plan talked about two steps to analysis: inductive and deductive. This makes sense for the study aim and was outlined well. However, we see little detail and evidence of stage 1 (the development of the inductive themes) in the results and instead see the outcome of the deductive analysis as we see data by COMB component. Again, this detail is good but I had expected to see details of step one and the themes identified there. This made me wonder why this stage was needed, could you have not coded the data deductively without this stage? It maybe that the themes were mentioned here e.g., 'uncertainty' and just needed some highlighting to show the steps.	The data was first analysed inductively, and then a deductive COM-B analysis applied to the themes that arose from the initial, inductive analysis, which is the focus for this paper. We took this approach to ensure a thorough and in-depth understanding of the presenting data before any deductive analyses are applied. As a team we felt that knowledge of the whole dataset is essential in order to be confident that a deductive approach is fully contextualised in and coherent with the rest of the data. The inductive, initial stages of analyses are key to being able to apply multiple, secondary deductive frameworks, with another planned using the NPT framework. The data presented in the Results section of this paper is based on the COM-B analysis.	To add context of our process, we have added the inductive themes to the beginning of the results section from line 182 and adapted the paragraph that follows: Initial inductive analysis identified the following overarching themes that influence behaviours related to the 6- to 8-week check: 'Training in paediatrics'; 'Knowing hip examination terminology and NIPE guidance'; 'Examination skills'; 'Experience in practice'; 'Designated baby clinics and time available'; 'Interaction with computer system'; 'Uncertainty in the physical hip examination'; 'Beliefs about hip examination as a priority among other

		examinations’; ‘Meaning and understanding of referral’; ‘Low threshold to refer’; and ‘Referrals need to be justified based on explicit criteria’. From this initial analysis, we deductively identified clear Capability, Opportunity and Motivation influences on GP behaviour in relation to DDH, which provided evidence for components which could be targeted in future interventions to improve referral pathways. This COM-B analysis is the focus of the current paper.
You mention PPI involvement which is great - what exactly did they do for this study? This detail would be great.	This study is part of a wider trial (Roposch et al., 2020) which is funded by an NIHR Programme Grant for Applied Health Research. There is a PPI group embedded in this programme of research; its members have helped to both secure the funding and advise on study materials such as study materials, approaches to interviews, recruitment etc. They also give feedback periodically on study conduct and findings.	We have reworded from line 171 to read: We developed the grant proposal, for which this study is part of, with input from GPs, carers of children with DDH and the founding director of ‘Steps’, a charity supporting patients with lower limb disorders. Our established patient and public involvement

		group periodically review and comment on study documents and findings, and have supported this current study with insight into the 6- to 8-week check process.
--	--	--

VERSION 2 – REVIEW

REVIEWER	Debbie Smith The University of Manchester I am interested in and designing an intervention for behaviour change more widely at the 6-8 week check. This follows on from work I have previously conducted (https://bjgp.org/content/68/669/e252).
REVIEW RETURNED	16-Feb-2021

GENERAL COMMENTS	Thank you for considering my previous comments regarding this paper. As I said previously, this is a great paper and very relevant and interesting topic. I have re-read the paper and my questions about the role of the observations, the checklist and the analysis procedure have all been addressed clearly. I have made one slight comment on the wording around the decision to stop recruitment but this is minor and is not essential. Thank you and I look forward to learning about the rest of the project.
---

VERSION 2 – AUTHOR RESPONSE

We thank you for your suggested revisions, we have taken them into account and made the applicable changes.